# The Role of Macrophages in Oocyte Donation Pregnancy: A Systematic Review

**DOI:** 10.3390/ijms21030939

**Published:** 2020-01-31

**Authors:** Xuezi Tian, Michael Eikmans, Marie-Louise van der Hoorn

**Affiliations:** 1Department of Gynecology and Obstetrics, Leiden University Medical Center, 2333 ZA Leiden, The Netherlands; X.Tian@lumc.nl; 2Department of Immunohematology and Blood Transfusion, Leiden University Medical Center, 2333 ZA Leiden, The Netherlands; M.Eikmans@lumc.nl

**Keywords:** Immunology, pregnancy, oocyte donation, macrophages

## Abstract

The embryo of an oocyte donation (OD) pregnancy is completely allogeneic to the mother, which leads to a more serious challenge for the maternal immune system to tolerize the fetus. It is thought that macrophages are essential in maintaining a healthy pregnancy, by acting in immunomodulation and spiral arterial remodeling. OD pregnancies represent an interesting model to study complex immunologic interactions between the fetus and the pregnant woman since the embryo is totally allogeneic compared to the mother. Here, we describe a narrative review on the role of macrophages and pregnancy and a systematic review was performed on the role of macrophages in OD pregnancies. Searches were made in different databases and the titles and abstracts were evaluated by three independent authors. In total, four articles were included on OD pregnancies and macrophages. Among these articles, some findings are conflicting between studies, indicating that more research is needed in this area. From current research, we could identify that there are multiple subtypes of macrophages, having diverse biological effects, and that the ratio between subtypes is altered during gestation and in aberrant pregnancy. The study of macrophages’ phenotypes and their functions in OD pregnancies might be beneficial to better understand the maternal-fetal tolerance system.

## 1. Introduction

During pregnancy, maternal immune cells circulate through the placenta and come into contact with cells from the fetus. In normally conceived pregnancy, the fetus is semi-allogeneic to the mother, expressing both maternal and paternal genes. Rejection is avoided and immunologic tolerance toward the fetus is achieved in uncomplicated pregnancies. In oocyte donation (OD) pregnancy, the fetus has an even greater antigenic dissimilarity compared to the mother, as both sets of genes are allogeneic: one from the oocyte donor and one from the father. This makes OD pregnancies an interesting model to study how the maternal immune system is guided into a state of tolerance during pregnancy.

Thirty-five years have passed since the first successful OD therapy [1], and OD is now applied world-wide. An international web-based survey showed that there were 14,890 OD cycles every year in only 161 units in the world, and the total amount was uncountable [2]. While there are some non-medical reasons for people to conceive through OD, the major indication is reproductive disorders, such as primary ovarian insufficiency, Turner syndrome, or maternal genetic diseases that parents do not want to transfer to the next generation [3]. However, OD pregnancy is an independent risk factor for some pregnancy complications [4], including preterm birth, low birth weight, pregnancy-induced hypertension, and preeclampsia [5,6,7]. A recent study showed that human leukocyte antigen (HLA) class II mismatches are associated with the development of preeclampsia in OD pregnancies [8]. This may suggest that antigenic dissimilarity affects the outcome of OD pregnancy through immune mechanisms. In addition, chemokines that are crucial for metabolic function in pregnancy, for example nitric oxide, might also affect the outcome of OD pregnancy [9].

From the beginning of pregnancy, an increase in immune cells is seen at the maternal-fetal interface, making them the main cell types encountered by invading extra villous trophoblasts (EVTs) [10]. The majority of immune cell types in the placenta are natural killer (NK) cells and macrophages, which consist of around 90% of all the leukocytes, leaving the remaining 10% primarily to T cells [11]. The maternal immune response can be triggered and regulated by human leukocyte antigen (HLA) of paternal origin from the fetus. The function of NK cells is regulated through the binding of receptors on NK cells to HLA molecules on EVTs. For example, inhibitory and activating killer cell immunoglobulin-like receptors (KIR) are present on the NK cell surface, both of which are able to interact with HLA-C allotypes [12]. As for T cells, there are two main types, CD4+ and CD8+ T cells, in the placenta bed. The main HLA molecule that T cells may react to is HLA-C, expressed on the fetal trophoblasts [13].

Whereas macrophages represent the second most abundant type of immune cell in the decidua [14], they maintain constant quantities during gestation, and increase even further at the time of parturition. This suggests that macrophages have a role in each trimester. The high number of macrophages around the invading trophoblasts and spiral arteries during implantation [15,16] implies a role for decidual macrophages [17] in trophoblast invasion, vascular remodeling, and placentation [18]. Monocytes get activated while they pass through the placenta, which might be due to their close contact with semi-allogeneic fetal villous syncytiotrophoblast [19]. In OD pregnancy, the maternal immune cells encounter the fully allogeneic fetus, which might activate the monocytes differently and more profoundly. Because of this heavier immune burden on the maternal-fetal interface in OD pregnancy, the presence of macrophages at the fetal-maternal interface surface may be more crucial. Higher amounts and stronger effects of macrophages are to be expected, and to maintain the invasion and tolerance of a fully allogenic fetus, their population and main functions might vary according to the layer in the placenta where they are situated.

In this review we will first give an overview of the role of macrophages before and during gestation in naturally conceived pregnancies. To get more insight into the role of macrophages in allogeneic OD pregnancies, we have performed a systematic review on the literature of macrophages in OD pregnancy.

### 1.1. Macrophages in Naturally Conceived Pregnancies

#### 1.1.1. Macrophages in the Immune System

Macrophages are large cells with multiple vacuoles and one nucleus. They are considered as phagocytes in the lineage of myeloid cells, cleaning cellular debris resulting from apoptosis or necrosis. However, in recent years, increasing amounts of evidence have generated some new theories about the origin and function of macrophages.

For many years, it was widely believed that monocytes travel from bone marrow into the blood circulation and get matured step by step while circulating in peripheral blood. In different stages of maturation, monocytes locate in peripheral tissues to become tissue-resident macrophages. However, recent evidence shows that during early gestation, from the yolk sac of three- to four-week-old human embryos, embryonic macrophages are already being produced [20]. For these primitive macrophages, some studies showed they do not rely on transcription factor Myb and bypass monocytic intermediates, suggesting that yolk sac macrophages grow independently [21]. In addition, once the fetal liver is operational, erythro-myeloid progenitors that seed in the fetal liver start to differentiate into tissue-resident macrophages [22]. Furthermore, a third pathway has been described that concerns generation of long-lived tissue-resident macrophages before birth, also present in fetal liver, but from hematopoietic stem cell-derived monocytes [23]. Therefore, adult tissue-resident macrophages seem to derive from different origins, in which none of them are from the bone marrow.

Rather than recruiting other macrophages from the blood, tissue-resident macrophages are able to renew themselves by local proliferation, under the influence of cytokines, for example, interleukin-4 (IL-4), which is released by T helper 2 (Th2) cells [24]. Another study pointed out that blood-derived monocytes might need to make up the number of tissue-resident macrophages, as they will lose their ability to proliferate with age [25].

It is commonly accepted that macrophages can polarize into two phenotypes, M1 (classically activated macrophages) and M2 (alternatively activated macrophages), but emerging evidence in recent years has shown there are far more than only two functional phenotypes in macrophages, which warrants a new theory to describe macrophage polarization. Several years ago, a theory that defined the differentiation of macrophages along with the line of a multidimensional color wheel was proposed, based on three major functions of macrophages, namely host defense, wound healing, and immune regulation [26]. Unfortunately, there is still no complete theory yet about macrophages’ differentiation.

#### 1.1.2. Macrophages in Endometrium before Pregnancy

Different leukocytes have been detected in the endometrium, each of them playing a role during different stages of the menstrual cycle, from endometrial breakdown to cellular proliferation [27]. Macrophages can be found in the endometrium throughout all the menstrual phases, where they regulate other cells through cell-cell interaction or cytokine production.

During the menstrual cycle, the number of macrophages in the endometrium keeps changing. These cells only represent a small proportion of endometrial cells in the proliferation phase but can gradually reach the peak amount of 6%–15% within the total population of endometrial cells during the menstruation phase [28]. While the total amount of endometrial cells is rising during the proliferation phase, the increasing percentage of leukocytes in non-proliferation phases may suggest that immune cells mainly do their job in these phases.

As the function and polarization of macrophages is regulated by the microenvironment [29], changing levels in hormones and cytokines might be the trigger to active endometrial macrophages. Estrogen and progesterone are the two main functional ovarian steroid hormones during the menstrual cycle. Estrogen can adjust macrophage recruitment into the uterus [30], and has the potential to regulate endometrial macrophages directly, as the estrogen receptor has been found on macrophages [31]. Progesterone might have an indirect influence since there is no progesterone receptor found on endometrial macrophages [32]. In addition, both estrogen and progesterone receptors can be found on peritoneal macrophages [33], suggesting that these hormones affect the uterine micro-immune environment through other macrophage-directed effects.

In each non-conception cycle, progesterone decreases in the late secretory phase. Endometrial perfusion is reduced with this progesterone withdrawal, so an inflammatory environment is built when facing this endometrial injury situation [34]. Some inflammatory pathways are activated in such an environment, such as the nuclear factor kappa-light-chain-enhancer of activated B cells (NF-κB) pathway, which leads to production of chemokines that recruit immune cells and that cause local activation of matrix metalloproteases (MMPs) [35]. MMPs are calcium-dependent zinc-containing proteinases, produced by macrophages, and they are considered to be essential in endometrial breakdown by unbinding peptide bonds of amino acids [36]. Besides helping in tissue break down, macrophages also prepare implantation before the menstruation phase. The cytokine macrophage inflammatory protein 1B (MIP-1B) is produced by macrophages in the secretory phase and maintains a pro-inflammatory environment to get the endometrium prepared for implantation [37].

As for the role in pathology of the endometrium, macrophages are still under debate. In endometriosis, the number of macrophages was found to be increased in the eutopic endometrium during the proliferation phase in one study [38] but decreased in another study [39]. The reason for these discrepant results may be variation in macrophage polarization and activation states. Taking this into account, different studies still showed diverse results. One study showed that the total amount of macrophages in the eutopic endometrium with endometriosis was increased. At the same time, the number of M2 was decreased in all phases, suggesting that the trend of macrophages polarizing to M1 might contribute to the cause of endometriosis [40]. However, a more recent study found that the eutopic endometrial homogenate from patients with endometriosis has the ability to differentiate macrophages from M1 to M2 [41]. This study also showed that M1 decreased in peritoneal washes from endometriosis patients, especially at a later stage of the disease (III and IV). Thus, it was hypothesized that this tendency leads to increased immune tolerance for ectopic endometrium, which could be the cause of endometriosis. Another study showed conflicting results with respect to peritoneal macrophages affecting the pathogenesis of endometriosis by creating a pro-inflammatory environment in the pelvic cavity [42].

A study in endometrioma also showed the trend that M1 were progressively decreased from stage I patients to stage IV patients, while M2 had an opposite tendency [43]. They explained this phenomenon by the notion that the microenvironment slightly changed from a pro-inflammatory one to a pro-fibrotic one when the disease became more severe. This suggests that the immune system represents a counterbalance in protecting the disease from deteriorating.

In adenomyosis, a similar concept applies. One study showed that M2 could induce the epithelial–mesenchymal transition (EMT) process [44], which gives the ability to endometrial cells to migrate into the myometrium. Another study claims that it is because macrophages are unable to polarize into M2 that cause adenomyosis [45].

#### 1.1.3. Macrophages during Pregnancy (Decidual Macrophages)

Macrophages are not only involved in the menstrual cycle, but also in the decidual layer during pregnancy. The origin of decidual macrophages is still largely unknown. Some articles suggest that decidual macrophages are mainly derived from monocytes in the peripheral blood, as an exception to tissue-resident macrophages derived from the yolk sac [46]. In contrast, another study provided evidence that macrophages in the human decidua are derived from proliferating endometrial macrophages that are already present in the endometrium before conception [47].

Regardless of the origin, macrophages constitute 20%–25% of the total leukocyte population in the decidua [48]. Unlike uterine NK cells, which are reduced in numbers with increasing gestational age, macrophages maintain a constant quantity over the course of pregnancy [14]. This may imply that macrophages are needed in each trimester.

Inflammatory conditions around the site of the conceptus are dynamic over the course of pregnancy. Implantation is the very beginning stage of pregnancy. At this stage, macrophages tend to secret cytokines to build a pro-inflammation environment in the secretory phase, which contains windows of implantation. Moreover, it is thought that the blastocyst needs to break through and damage the endometrial tissue in order to invade [49]. Thus, it seems beneficial to have a pro-inflammatory environment in the preimplantation phase rather than an anti-inflammatory environment.

A study in mice showed that uterine cells synthesize granulocyte macrophage colony-stimulating factor (GM-CSF) within the hour of natural insemination [50]. Another mouse study showed the potential of seminal plasma to induce inflammatory cytokines, such as IL-1b and tumor necrosis factor (TNF), when the embryo was still traveling in the oviduct [51]. They also found infiltration of macrophages in the uterus in this very early stage, and the macrophages were mainly skewed to M1. However, another study showed that the total quantity of macrophages in the endometrium was decreased from the time of mating to implantation [52]. More research is needed to investigate the function of macrophages in the preimplantation stage.

As soon as the blastocyst implants in the endometrium, the interaction between maternal cells and fetal cells is initiated. After blastocyst attaching the uterine lining, trophoblast will proliferate into two layers, of which the outer layer, also called syncytiotrophoblast, forms the first contact with maternal cells, to later play a role in forming a barrier for exchange of nutrients and gases. The inner layer (cytotrophoblast) will continue its proliferation into EVTs, some of which will invade into the decidua and surround the uterine spiral arteries [53]. Mainly NK cells and macrophages are recruited in that particular location, even before the presence of trophoblast cells, suggesting that they play an essential role in the first step of spiral arterial remodeling [46]. Macrophages might also act as phagocytes to engulf the apoptotic cell, not only preventing proinflammatory reactions in this area, but also regulating the extent of fetal cells invading into the uterine wall [54,55].

During pregnancy, trophoblast cells only express a limited number of classical HLA molecules to maintain immune evasion toward maternal immune cells. Only three HLA subtypes have been detected on trophoblasts, namely HLA-C, HLA-G, and HLA-E [56]. By binding to HLA-G with Ig-like transcript (ILT) 2 and ILT4 receptors on their surface, macrophages may exert inhibitory and regulatory effects [57] to aid in immune tolerance.

Macrophages can provide multiple functions during pregnancy, which is generally related to their phenotype. Broadly, macrophages in the decidua can be distinguished into two phenotypes, M1 and M2. To detect and distinguish these two phenotypes, several cell markers and cytokines can be used. The primary cell marker to distinguish decidual macrophages is cluster of differentiation (CD) 14, which is mainly the marker of monocytes that have the potential to differentiate into macrophages, irrespective of whether they are pro- or anti-inflammatory [58]. CD68 is a more common marker to describe macrophages in many different tissues [59], and while CD163 is used to detect M2 [60], costimulatory molecules CD80 and CD86 rather represent M1 in a pro-inflammatory environment [61]. Moreover, GM-CSF from the endometrium can activate macrophages into M1-like macrophages that produce IL-12 and tumor necrosis factor alpha (TNF-α) as a pro-inflammatory cytokine. Trophoblast-derived macrophage colony-stimulating factor (M-CSF) has the ability to differentiate macrophages into M2, which produce IL-10 and transforming growth factor beta (TGF-β) as an immunoregulatory cytokine [62,63,64]. Currently, more and more cell markers are found to distinguish subtypes of macrophages. CD11c is considered to be an important marker to distinguish different subsets of decidual macrophages: CD11c^HI^ macrophages act more as immune regulators and CD11c^LO^ macrophages are more involved in phagocytic processes [65]. In addition, there are two subsets of decidual macrophages in early pregnancy that have different expression of CD209, and that do not fit in the M1/M2 phenotypes model [66]. Thus, the division of M1 and M2 seems to be inadequate for describing the total spectrum of decidual macrophages. In line with this, decidual macrophages have been distinguished in to five subsets: M1, M2a, M2b, M2c, and M2d, in which each subset is different in function and cytokines’ production [17].

At term, when approaching delivery, the differentiation of macrophages will slightly change again. Macrophages in this term tend to have a stimulatory capacity toward other immune cells, by secreting more IL-12 rather than IL-10 or TGF-β [67].

Pregnancy complications might be associated with an abnormal quantity of macrophages in the decidua. Some studies showed that there is an increased number of total macrophages in the decidua and a reduced number of M2 surrounding the spiral arterials of patients with preeclampsia in the early stage of pregnancy [68,69]. However, it is hard to define whether this phenomenon is the cause or the result of preeclampsia. Not only the number but also the function of macrophages in preeclampsia needs to be defined.

## 2. Macrophages and OD Pregnancy (Systematic Review)

In pregnancies conceived after OD, the fetus is fully allogeneic to the mother. Therefore, OD pregnancies are considered to be more immunologically challenging than naturally conceived pregnancies. In OD pregnancies, the maternal immune cells encounter the fully allogeneic fetus, which might activate the monocytes differently and more profoundly compared to naturally conceived pregnancies. Despite a continued increase in the number of OD pregnancies, relatively little is known about the underlying biology. The aim of this systematic review is to assess the role of macrophages in human OD pregnancy compared to naturally conceived or with assisted reproductive techniques.

### 2.1. Methods

#### 2.1.1. Search Strategy and Study Selection

The literature search was conducted by an experienced librarian, J. W. Schoones, using the following databases: PubMed, Embase, Web of Science, COCHRANE Library, Emcare, Academic Search Premier, Eligibility criteria. PRISMA guidelines for systematic reviews was followed (Appendix A). Original articles were searched in all these database. We used “oocyte donation”, “egg donation”, “ed pregnancies”, “od pregnancies”, “allogeneic pregnancy”, in combination with: “macrophages”, ”myeloid cells”, “phagocytes”, “epithelioid cells”, “monocytes”, “myeloid-derived suppressor cells”, “ myeloid progenitor cells”, and “CD14”, “CD68”, and “CD163” as our key words. The full electronic search strategy for the databases is shown in Appendix A. There was no restriction about language or time. The literature search was performed on 30 September 2019. The titles and abstracts were evaluated by three independent authors (M.L.P. van der Hoorn, M. Eikmans, X. Tian), and any uncertainty and difference was resolved by discussion. Inclusion criteria were original studies in human with relevance to the aim of our review. Reviews and systematic reviews, case reports, and letters were excluded.

#### 2.1.2. Risk of Bias Assessment

The quality of studies was assessed using the Newcastle-Ottawa Scale (NOS) (http://www.ohri.ca/programs/clinical_epidemiology/nosgen.pdf). All studies were independently rated by three reviewers (M.L.P. van der Hoorn, M. Eikmans, X. Tian). The NOS for case-control studies refers to three aspects: selection, comparability, and exposure criteria. The NOS for cohort studies refers to three aspects: selection, comparability, and outcome criteria. The total score represented the sum of all aspects. This score was used as a relative measure of data quality, of which 7 or higher were considered high-quality studies, and 5 to 6 were moderate quality [70]. The results are shown in the Appendix A.

### 2.2. Results

Details of the study selection process are shown in the flow diagram (Figure 1). The systematic search led to the retrieval of a total of 291 articles. After removing duplicates, 138 articles remained for first-stage screening. Next, by reviewing titles and abstracts, 130 articles were excluded. No additional articles were included by checking the references. 

Eight articles were identified to assess the full text for eligibility, four of which were animal studies. Therefore, in the end, four articles met all the inclusion criteria and were included in this systematic review.

Table 1 shows the main characteristics of the four studies included in this systematic review. All studies were performed after 2010. One out of the four studies did not have a naturally conceived control group [71]. In this study, they compared OD cases with and without a distinct lesion. All studies, except for Nakabayshi et al. [72], included vaginal deliveries and caesarean sections. Two studies [71,73] collected placentas of OD pregnancies. Martinez-Varea et al. [74]. investigated cytokines in the peripheral blood throughout pregnancy. All other studies looked at immunological components after pregnancy. Nakabayashi et al. [72] collected biopsies from the placental implantation side, at the time of caesarean section. In addition, they have collected the uterus after hysterectomy after cesarean section because of atonic bleeding or placenta accrete in three cases.

### 2.3. Discussion

All four studies included in this systematic review showed an aberrant macrophage function in OD pregnancies compared to in vitro fertilization (IVF) or naturally conceived (NC) pregnancies. The main results of the studies are summarized in Figure 2.

One of the four included articles investigated the cytokine and chemokine profile in peripheral blood of women pregnant after OD, compared to IVF or NC pregnancies [74]. Interestingly, most cytokines and chemokines characterized in this article (GM-CSF, IFNγ, IL10, IL17A, IL1β, IL2, IL4, IL5, IL6) had a similar pattern during the period of pregnancy: the cytokine levels rapidly rose in the second trimester and remained high in the third trimester.

The cytokine level during gestational age of some cytokines acted differently compared to the majority and to each other, of which stromal cell-derived factor-1alpha (SDF1α), TNF-α, and IL-8 can be related to macrophages functions [74]. TNF-α can be released by M1-type macrophages, and it can also be expressed as a proinflammation cytokine by other immune cells, such as T cells in pregnancy [75]. Therefore, it seems reasonable that TNF-α, raised in levels during the first trimester, which decreased during the second trimester and then remained low afterwards, helped in proper implantation first and then contributed to immune tolerance. IL-8 also had a dynamic pattern, and its levels were constantly raised during the whole gestation in the peripheral blood. The amount of IL-8 in OD and IVF pregnancy was higher than that in NC pregnancy during each trimester. IL-8 is referred to as a cytokine that regulates endothelial cell proliferation and angiogenesis [75], and in another article, it was considered to recruit inflammatory cells [76]. This might reveal that IL-8 plays a role in diverse aspects during pregnancy. But as many cell types may produce IL-8, including macrophages, neutrophils, and epithelial cells, the exact relationship between dynamic IL-8 patterns and macrophages is still uncertain.

The maternal humoral immune response throughout pregnancy might be similar in pregnancies with a totally allogeneic fetus compared with those with a semi allogeneic fetus (NC and IVF). However, in peripheral blood of OD pregnancy, the level of SDF1α is significantly lower in the third trimester than in the two other pregnancy groups [74]. It is thought that the lower SDF1α levels in the third trimester of OD pregnancies, compared to gestations with autologous oocytes, may contribute to maintaining these pregnancies with a fully allogeneic fetus. SDF1α is considered to be a cytokine involved in tissue repair and transplantation tolerance, since a high expression of SDF1α is associated with allograft rejection [77].

The relation of SDF1α to macrophages is described in other research [78]. A renal study in mice showed that the timing in upregulation and the location of infiltration of macrophages is similar to those of SDF1α, suggesting there might be an interaction between macrophages and SDF1α [79]. Another study found that human multiple myeloma cells could induce macrophage polarization to M2, through an axis that involves SDF1α [80]. However, since SDF1α has also been associated with allograft rejection [77], its function is still uncertain and the relation between SDF1α and macrophages in OD needs to be further investigated.

The investigation of the immunological mechanisms at the fetal-maternal interface might give more insight into the processes leading to the acceptance of the fetal allograft. However, obtaining local samples from this gestational age is logistically difficult. The other three studies included in this systematic review concentrated on the investigation of the fetal-maternal interface in term placenta and decidua.

Gundogan et al. [73] performed research on the macroscopy and microscopy of OD placentas. More pathological lesions in placentas of OD pregnancies than in non-donor IVF were described earlier [81]. Gundogan et al. [73] focused on the immune responses in the placenta which might have caused these pathological changes. They have found significantly higher rates of CD45+ (leukocytes), CD4+ (T cells), and CD56+ (NK cells) in placenta of OD pregnancies compared to non-donor IVF pregnancies. Macrophages were not investigated in this study. Two pathological changes in placenta were compared, chronic deciduitis and villitis of unknown etiology (VUE). The latter is mainly diagnosed by the presence of chronic inflammatory cells like T cells or macrophages, though they might come from different origin (T cells from the mother and macrophages from the fetus) [82]. No differences were found in the presence of VUE between OD and NC pregnancies. However, there was a significantly higher presence of chronic deciduitis in OD placentas compared to control placentas.

Nakabayashi et al. [72] investigated the immune mechanisms of OD in the decidual basalis. Schonkeren et al. [71] focused on the chorionic plate of placentas of OD pregnancies.

Nakabayashi et al. [72] showed that CD68+ cells were significantly decreased in pregnancies complicated by preeclampsia of non-OD group compared to pregnancies without preeclampsia. A significant decrease of these cells was also found in the OD group, with or without preeclampsia, compared to the non-OD group. In contrast, the study of Schonkeren et al. in the chorionic plate of placentas from OD pregnancies showed that CD14+ macrophages were increased in uncomplicated OD pregnancies compared to pregnancies complicated by preeclampsia. Nakabayashi et al. [72] put forward the hypothesis that macrophages infiltrate from decidua basalis to the basal plate, to explain the decreased numbers of macrophages in the decidua basalis and the increased numbers in the chorionic plate. However, the underlying reason still needs to be defined.

In addition, Nakabayashi et al. [72] used the ratio of p62+ cells to cytokeratin 7 (CK7)+ cells in the decidual basalis to describe the impaired autophagy of EVTs. p62 is a ubiquitin-binding protein that could be degraded by autophagy, the increase of p62 in CK7+ EVT cells might be due to the impaired autophagosome, which might inhibit the invasion of EVTs and vascular remodeling [83]. In this study, the ratio of p62+/CK7+ was significantly more increased in normotensive OD pregnancies than in normotensive NC pregnancies, indicating that impaired autophagy of EVTs is higher in OD pregnancies. Since macrophages have the ability to interact with EVTs and remodel spiral arteries, we speculate that the ratio of p62+/CK7+ reflects the effect of macrophages in OD pregnancy to some extent.

Pathological changes were found in both studies [71,72]. Nakabayashi et al. [72] found impaired vascular remodeling in the decidual basalis of OD pregnancy, and showed that OD pregnancy is an independent risk factor for preeclampsia. Schonkeren et al. [71] found a distinct lesion in 10 out of 26 chorionic plates of OD pregnancies, and observed a significantly lower incidence of preeclampsia in OD patients with this kind of lesion compared to women with OD pregnancy having this lesion (0% versus 45.5%, respectively). The lesion might be the consequence of maternal-fetal immune interaction, as some specific immune mechanisms could explain the occurrence of preeclampsia [84]. The inflammatory lesion contained high numbers of CD14+ and CD163+ cells, suggesting the abundance of type 2 macrophages. Combining the results, the conclusion can be drawn that type 2 macrophages initiate an immune mechanism to protect patients from preeclampsia. Moreover, the macrophages were found to be of maternal origin, and the women with a lesion in the placenta more often had a child with the HLA-C2 type. This suggests that fetal-HLA-C interactions mediate protective effects against preeclampsia, by raising the amount of type 2 macrophages in the chorionic plate. Further research needs to be performed to elucidate the role of specific maternal-fetal HLA mismatches in OD pregnancies and the association with pregnancy outcome.

It is valuable to investigate the macrophages present at the fetal-maternal interface in vivo over the course of pregnancy, as three out of four studies discussed in this review investigated macrophages after delivery and one study investigated macrophages in peripheral maternal blood.

## 3. Conclusions

Macrophages play an essential role in the immune system, and they have a role in maintaining healthy pregnancy. The study of macrophage phenotypes and its functions in an OD situation will help to better understand immune mechanisms at the maternal-fetal interface. Unravelling how the fetal tissue is accepted in pregnancy will provide information for the field of fertility, obstetrics, and transplantation.

From current research, we could summarize that there are multiple subtypes of macrophages, having diverse biological effects. We have also seen that the ratio between macrophages’ subtypes is altered both during gestation and in aberrant pregnancy. To better understand the ultimate immunological paradox in OD and to build a complete pathway of how macrophages maintain successful OD pregnancy, further investigations are needed to address different macrophages’ subsets in different locations of the placenta, their functionality, and their interaction with placental and immune cells.

Moreover, it is useful to analyze relationships between HLA mismatches, placenta lesions, and macrophage loads, as these factors could be related to macrophage function. Confirmation that a higher level of HLA incompatibility results in differential immunological responses would imply that oocyte donors may be selected on the basis of the degree of HLA matching, leading to a higher number of clinically successful OD pregnancies. Furthermore, elucidation of the functional role of myeloid cells in the pathophysiology of pregnancy disorders may, in the future, lead to using their phenotypic characteristics as a biomarker of pregnancy outcome and/or an indicator of stratification for therapeutic intervention.

## Figures and Tables

**Figure 1 ijms-21-00939-f001:**
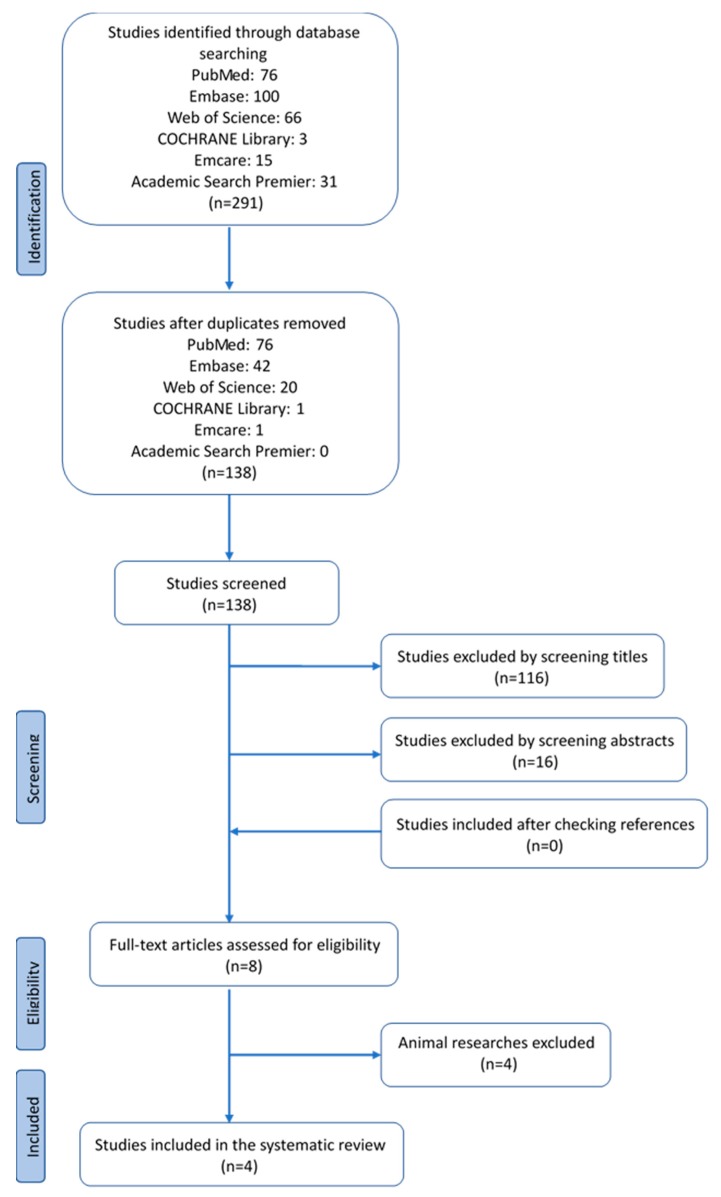
Four articles met all inclusion criteria and were included in the systematic review.

**Figure 2 ijms-21-00939-f002:**
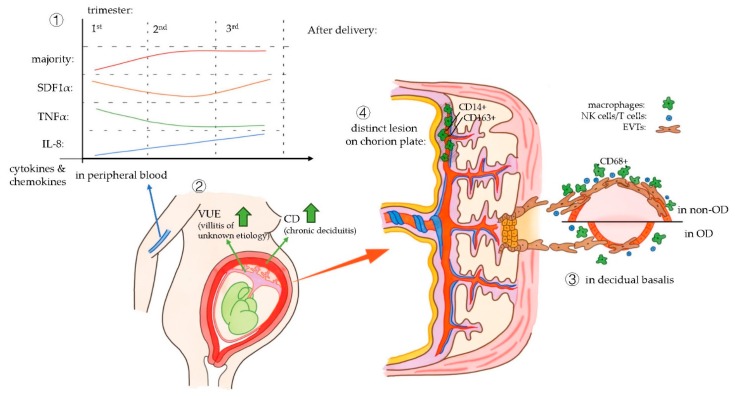
Main outcomes of four articles which are included in this systematic review.: ① The results of the study by Martinez-Varea et al., in peripheral blood. ② The results of the study by Gundogan et al. on placenta pathology findings in OD pregnancies. ③ The study by Nakabayashi et al. with the results on impaired vascular remodeling in the decidual basalis of OD pregnancies. ④ The study by Schonkeren et al. showed a distinct pathological lesion in the chorionic plate of OD placentas.

**Table 1 ijms-21-00939-t001:** Overview of the articles selected in this systematic review.

First Author	Year	Journal	Study	Samples	Delivery	Material	Methods	Main Outcomes	
Martinez-Varea et al.	2015	Journal of Immunology Research	Prospective longitudinal study	25 OD 25 IVF 25 NC	Vaginal and c-section	Peripheral maternal blood at different time points during gestational age	Cytokine analysis (Luminex)	three study groups displayed similar cytokine and chemokine patterns throughout pregnancy	OD pregnancies showed lower SDF1α levels in the third trimester compared with NC and IVF pregnancies
Gundogan et al.	2010	Fertility and sterility	Retrospective case control	20 OD 33 non donor IVF	Vaginal and c-section	Placenta macro and microscopy. Perinatal data	IHC Pathological investigation	Significant histological and immunohistochemical differences between the placentas of OD and nondonor IVF pregnancies	Representation of a host versus graft rejection-like phenomenon in OD
Nakabayashi et al.	2016	Journal of reproductive Immunology	Case control	19 OD 22 NC 7 IVF	All c-sections	Decidua basalis form placental site uterus. Implantation site biopsy. Uterus after hysterectomy (*n* = 3).	IHC	Frequencies in normotensive OD pregnancies or preeclamptic cases in OD pregnancies were similar to those in preeclamptic cases in NC/IVF	Numbers of decidual CD3+ T cells, CD8+ T cells, CD4+ T cells, Foxp3+ T cells, CD56+ NK cells, and CD68+ macrophages were significantly decreased in the decidua basalis of OD patients compared with those in normal pregnant subjects impaired autophagy of EVTs was higher in normotensive OD pregnancy than in normotensive NC pregnancy
Schonkeren et al.	2012	PLOSone	Prospective cohort study	26 OD placentas	Vaginal and c-section	Placenta macro and microscopy. Perinatal data	IHC FISH HLA typing KIR genotyping	Distinct lesion only in OD placentas, and with this lesion, the incidence of PE is low	Expression of CD14+ and CD163+ cells in the lesions

OD: oocyte donation; IVF: in vitro fertilization; NC: naturally conceived; IHC: immunohistochemistry; HLA: human leukocyte antigen; FISH: fluorescence in situ hybridization

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
