# Peer review of "The Role of Macrophages in Oocyte Donation Pregnancy: A Systematic Review"

_ijms, 2020, doi:10.3390/ijms21030939_

Round 1

Reviewer 1 Report

I read with great interest the Manuscript titled “The role of macrophages in oocyte donation 2 pregnancy: a systematic review” (ijms-693442), which falls within the aim of the International Journal of Molecular Sciences.      

In my honest opinion, the topic is interesting enough to attract the readers’ attention. The methodology is accurate and conclusions are supported by the literature data reported. Nevertheless, the authors should clarify some points and improve the discussion citing relevant and novel key articles about the topic.

Authors should consider the following recommendations:

Was this systematic review registered in PROSPERO? I did not find any information about this point. The Authors did not report the results of the assessment of the risk of bias. I think this is very important information to report so I suggest that the authors include these results in their manuscript. It is possible to report the risk of bias assessment as a table. I suggest to stress, at least briefly, the novel pieces of evidence regarding the role of M1 and M2 Macrophages in endometriosis (refer to PMID: 31663401; PMID: 30276219) and discuss the potential correlation with the same alterations in eutopic endometrium, whose environment is known to orchestrate the maternal-fetal interface. I appreciated the detailed description of factors regulating immune tolerance in the decidual tissue first, and then in the placenta. I would suggest adding a few lines to further discuss how changes in key immune populations may switch the response and lead towards obstetric diseases (refer to PMID: 28282763; PMID: 28466013).

Author Response

We thank the reviewer for this comment. This review was not registered in PROSPERO. Before we started writing the review, we checked PROSPERO in order to assess whether the idea of the review was already claimed by another group, which was not the case. The aim of PROSPERO to provide a comprehensive listing of systematic reviews registered at inception to help avoid duplication and since we have had the experience with IJMS having a very fast processing time, we would not expect that duplication has happened. In addition, to date there has no review been registered with this subject.

We agree on the importance of the assessment of the risk of bias. Therefore, we have added a paragraph in the method section on the risk of bias assessment and we have made a table on the risk of bias as a supplementary table. 

We have added the role of M1 and M2 macrophages in endometriosis as the reviewer suggested. (see Page 3. Line 144-160)

Our review focuses on macrophages and their role in relation to obstetric diseases (see page5. line230-235). We did not include other immune populations in our review since to our opinion this is beyond the scope of the review.

Reviewer 2 Report

This is a review on the role of macrophages in oocyte donation (OD) programs, a delicate moment in which the maternal immune system should tolerate an allogenic fetus.

The review starts with an accurate and deep overview of the role of macrophages before and during gestation in naturally conceived pregnancies to then focus on the systematic review. Unfortunately, from 138 studied screened only 4 (2010-2016) were adherent to inclusion criteria.

The topic is of interest and a scarse literature is available on this specific matter; however, before acceptance I suggest to the AA to check an eventual content overlapping in the Introduction with a recent review (Magatti et al., 2019. Cells; 8(11)).

Moreover, since the groups already published papers on different aspects of this topic (not cited here), including two prospective studies:

1: van Bentem K, Bos M, van der Keur C, Brand-Schaaf SH, Haasnoot GW, Roelen DL, Eikmans M, Heidt S, Claas FHJ, Lashley EELO, van der Hoorn MLP. The development of preeclampsia in oocyte donation pregnancies is related to the number of fetal-maternal HLA class II mismatches. J Reprod Immunol. 2019;137:103074.
2: du Fossé N, van der Hoorn ML, Eikmans M, Heidt S, le Cessie S, Mulders A, van Lith J, Lashley E. Evaluating the role of paternal factors in aetiology and prognosis of recurrent pregnancy loss: study protocol for a hospital-based multicentre case-control study and cohort study (REMI III project). BMJ Open. 2019;9(11):e033095.
3: van Bentem K, Lashley E, Bos M, Eikmans M, Heidt S, Claas F, le Cessie S, van der Hoorn ML. Relating the number of human leucocytes antigen mismatches to pregnancy complications in oocyte donation pregnancies: study protocol for a prospective multicentre cohort study (DONOR study). BMJ Open. 2019;9(7):e027469.
4: Bos M, Schoots MH, Fernandez BO, Mikus-Lelinska M, Lau LC, Eikmans M, van Goor H, Gordijn SJ, Pasch A, Feelisch M, van der Hoorn MP. Reactive Species Interactome Alterations in Oocyte Donation Pregnancies in the Absence and Presence of Pre-Eclampsia. Int J Mol Sci. 2019;20(5). pii: E1150.

it is suggested to make a comment on those aspects presented in this review (as HLA, for example)

Author Response

We thank the reviewer for this comment. We compared our review with the review from Magatti et al. While they focused on the effect of MSC on macrophages, we focused more on the effects of functionality of the macrophages. Thus, we described macrophages in a different way, and our review does not overlap with their review.

We thank the reviewer for the suggestion to make a comment on the prospective studies. Therefore, we have added a short part to the Introduction to comment on the prospective studies.  (see Page1. Line44-48).

Reviewer 3 Report

please see attachment.

Author Response

We thank the reviewer for the useful comments.